# NIR Instruments and Prediction Methods for Rapid Access to Grain Protein Content in Multiple Cereals

**DOI:** 10.3390/s22103710

**Published:** 2022-05-13

**Authors:** Keerthi Chadalavada, Krithika Anbazhagan, Adama Ndour, Sunita Choudhary, William Palmer, Jamie R. Flynn, Srikanth Mallayee, Sharada Pothu, Kodukula Venkata Subrahamanya Vara Prasad, Padmakumar Varijakshapanikar, Chris S. Jones, Jana Kholová

**Affiliations:** 1Crop Physiology & Modeling, International Crops Research Institute for Semi-Arid Tropics, Patancheru, Hyderabad 502 324, India; keerthichadalawada@gmail.com (K.C.); a.krithika@cgiar.org (K.A.); s.choudhary@cgiar.org (S.C.); srikanthmallayee@gmail.com (S.M.); 2Department of Botany, Bharathidasan University, Tiruchirappalli 620 024, India; 3Crop Physiology & Modeling, International Crops Research Institute for Semi-Arid Tropics, Bamako BP 320, Mali; a.ndour@cgiar.org; 4Hone, Newcastle, NSW 2300, Australia; william@honeag.com (W.P.); jamie@honeag.com (J.R.F.); 5South Asia Regional Center, International Livestock Research Institute, Patancheru 502 324, India; p.sharada@cgiar.org (S.P.); k.v.prasad@cgiar.org (K.V.S.V.P.); v.padmakumar@cgiar.org (P.V.); 6Feed and Forage Development, International Livestock Research Institute, Addis Ababa P.O. Box 5689, Ethiopia; c.s.jones@cgiar.org; 7Department of Information Technologies, Faculty of Economics and Management, Czech University of Life Sciences Prague, 165 00 Prague, Czech Republic

**Keywords:** cereals, protein, near-infrared spectroscopy (NIRS), prediction methods, winISI, Hone Create, Convolution Neural Network (CNN)

## Abstract

Achieving global goals for sustainable nutrition, health, and wellbeing will depend on delivering enhanced diets to humankind. This will require instantaneous access to information on food-source quality at key points of agri-food systems. Although laboratory analysis and benchtop NIR spectrometers are regularly used to quantify grain quality, these do not suit all end users, for example, stakeholders in decentralized agri-food chains that are typical in emerging economies. Therefore, we explored benchtop and portable NIR instruments, and the methods that might aid these particular end uses. For this purpose, we generated NIR spectra for 328 grain samples from multiple cereals (finger millet, foxtail millet, maize, pearl millet, and sorghum) with a standard benchtop NIR spectrometer (DS2500, FOSS) and a novel portable NIR-based instrument (HL-EVT5, Hone). We explored classical deterministic methods (via winISI, FOSS), novel machine learning (ML)-driven methods (via Hone Create, Hone), and a convolutional neural network (CNN)-based method for building the calibrations to predict grain protein out of the NIR spectra. All of the tested methods enabled us to build relevant calibrations out of both types of spectra (i.e., R^2^ ≥ 0.90, RMSE ≤ 0.91, RPD ≥ 3.08). Generally, the calibration methods integrating the ML techniques tended to enhance the prediction capacity of the model. We also documented that the prediction of grain protein content based on the NIR spectra generated using the novel portable instrument (HL-EVT5, Hone) was highly relevant for quantitative protein predictions (R^2^ = 0.91, RMSE = 0.97, RPD = 3.48). Thus, the presented findings lay the foundations for the expanded use of NIR spectroscopy in agricultural research, development, and trade.

## 1. Introduction

Near-infrared spectroscopy (NIRS) is a non-destructive method that is widely used to predict the organic compounds of grain materials based on electromagnetic wave interactions. This technology offers time- and cost-effective options to analyze grain quality parameters [1,2,3,4]. While several companies offer the standard benchtop NIR spectrometers (such as the FOSS-DS2500 flour analyzer, [5], Bruker’s Tango FT-NIR spectrometer [6], Perten-IM9520 [7]), within the last decade, the market has offered several options for portable NIR instruments as well [8,9,10]. Examples include the MicroNIR OnSite-W from VIAVI Solutions [11], the DLP NIRScan^TM^ Nano EVM spectrophotometer from Texas Instruments’ DMD™ [12], the MEMS spectrometer from Fraunhofer [13], and the Hone Lab Red from Hone [14]. While many benchtop NIR instruments are already used for standard applications across the agri-food sector, portable instruments are not regularly used [15,16,17]. This is because accurate NIRS-based predictions require, among others, quality instrumentation to generate spectra, as well as robust prediction methods, and both can be problematic for some of the portable NIRS instruments [18,19,20,21,22,23].

There are many software packages enabling the prediction of material composition from the NIR spectra, which build on classical deterministic methods such as principal component analysis (PCA), partial least squares regression (PLSR), and multiple linear regression (MLR) [20,23,24,25,26,27,28,29,30,31,32,33]. Recent software programs offer machine learning (ML)-based methods—such as random forest, support vector machine regression, or stacked ensembles. However, more complex ML-based methods still require user configuration or custom software builds [34,35,36,37,38,39,40,41]. ML-based methods, in particular, are gaining a lot of attention, as these may offer specific advantages for applications where feature prediction is required from imperfect spectra or spectra derived from a range of materials. This is because ML-algorithms have the intrinsic capacity to identify common patterns in diverse data information, from which the required algorithms are built [38].

In the case of the cereal-based food and feed industry [25,42,43], benchtop NIRS systems with calibrations based on single-cereal species are routinely utilized—rice [26,28,30,32,36,40,44,45,46,47,48], sorghum [43,49,50], wheat [27,51,52,53,54,55], corn [31,53,56,57,58,59,60], and barley [53,61,62,63]. These typically use a large number of single-species samples from different environments representing a relevant range of target trait variability [3,52,64,65,66,67,68,69]. With rising global attention on food quality, minor cereals (such as sorghum, fonio, teff, and millets) are being explored and promoted [70,71,72,73]. These minor cereals are important sources of human and livestock diets that significantly influence their nutritional status and health [74,75,76], especially in the tropics [77,78,79,80]. However, for the industrial use of these cereals, rapid access to their grain components is required, and this is currently limited. At the same time, the trait variability within the minor species may be another significant bottleneck constraining the development of robust calibrations. Furthermore, even if large variable datasets were available for these minor crops, the development and maintenance of separate calibrations for each of these cereal species may prove to be time- and cost-inefficient. Therefore, a multi-cereal species calibration could be a convenient alternative. Most of the reported multi-crop calibrations focus on forage and feed analysis [81,82,83,84,85,86,87]. Very few reports have documented robust calibration models for grain across multiple species using the classical statistical modelling methods. A few authors have argued that ML-based prediction approaches could improve the reliability of multi-species calibrations [88].

So far, there are not many examples where NIRS-based systems have been used for the quality assessment of minor cereal grains [89,90,91,92]. In our case, we were interested in evaluating some of the novel instruments and methods. While the standard benchtop NIRS instruments are routinely used and certified for the assessment of grain quality (e.g., wheat in Australia) [93,94], the relevance of the upcoming generation of portable NIR instruments is not commonly standardized or documented [95]. Therefore, we aimed to evaluate the efficiency of several NIRS technology options (instruments and software) for the prediction of grain protein content in multiple cereal species. The emerging NIRS technology (portable instruments and ML-based prediction methods) were included in the study to test whether these could suit the specific needs of de-centralized agri-food value chains. The specific objectives of the study were to:(i)Compare the NIR spectra produced by benchtop (DS2500, FOSS) and portable (HL-EVT5, Hone) instruments, and their suitability for predicting grain protein in multiple cereal species;(ii)Assess the suitability of different model-building methods (via winISI, FOSS; Hone Create, Hone; and a customized CNN-based method) to predict protein content in multiple cereals using two types of NIR spectra;(iii)Ascertain the predictions made using multiple instrument–method combinations (i.e., FOSS-DS2500–winISI; FOSS-DS2500–Hone Create; FOSS-DS2500–CNN; HL-EVT5–Hone Create; and HL-EVT5–CNN) and discuss the suitability of their applications (for example, in decentralized breeding programs and markets).

## 2. Materials and Methods

The overall methodology used in the study is summarized in Figure 1. Briefly, 328 grain samples from five cereals were used for building NIR-based multiple cereal prediction models for estimating grain protein content. For this two NIR instruments and three calibration model building methods were explored. Also, the relevance of these instrument–method combinations were assessed using linear regression and goodness of fit parameters. Each of the steps are described in the sections below.

### 2.1. Plant Material

A total of 328 grain samples from 5 cereal species—154 genotypes of sorghum (*Sorghum bicolor* (L.) Moench), 125 genotypes of pearl millet (*Cenchrus americanus* (L.) Morrone), 20 genotypes of finger millet (*Eleusine coracana* Gaertn.), 19 genotypes of foxtail millet (*Setaria italica* (L.) P. Beauvois), and 10 genotypes of maize samples (*Zea mays* L.; details in Appendix A) were used in the study. The maize cultivars were obtained from the maize improvement program of the International Maize and Wheat Improvement Center (CIMMYT), and the remaining material were from the genebank repository of ICRISAT (Patancheru, India) [96] and the ICRISAT crop improvement programs. The subset of 154 sorghum samples included 4 races (bicolor, caudatum, durra, and guinea) originating from Burkina Faso, Cameroon, Ethiopia, India, Lesotho, Mali, Nigeria, and USA [97,98,99,100]. The subset of 125 pearl millet samples used in the study comprised 100 lines from the pearl millet inbred germplasm association panel (PMiGAP) [101] and 25 elite cultivars of Asian and African origin. Samples of 20 finger millet genotypes originating from India, Kenya, Malawi, Senegal, Uganda, and Zimbabwe, and 19 foxtail millet genotypes originating from China, India, Iran, Pakistan, Russia, and USA were used in the study [96].

### 2.2. Sample Collection and Preparation

The crops were raised on alfisol soil with recommended management practices [102] under irrigated conditions at the ICRISAT campus (Patancheru, India, 17.53° N, 78.27° E, 545 m.a.s.l) during the post-rainy season (October 2018–January 2019). The panicles from physiologically mature plants were harvested and manually threshed. Grains of each genotype were pooled, cleaned, and ground to flour of <1 mm particle size, using a CM 290 Cemotech™ laboratory grinder (FOSS, Hillerød, Denmark). The flour samples were then stored in 50 mL conical polypropylene Falcon tubes at 4 °C until laboratory analysis (see Section 2.3) and scanning with NIR instruments (see Section 2.4).

### 2.3. Laboratory Analysis of Grain Protein Content (“Ground Truth” Dataset)

The flour samples were dried at 130 °C for 2 h in an oven and cooled to room temperature prior to chemical analysis. Standard AOAC (2000) protocols [103] were followed to estimate moisture (AOAC 925.10) and total nitrogen content (N%; Kjeldahl method, AOAC 2001.11) in each sample (i.e., for each genotype separately; see Section 2.1). The total protein content was then calculated using the generically agreed consensus for N content conversion into protein [104,105]; i.e., by multiplying the nitrogen content with a protein conversion factor of 6.25 (Equation (1)).
(1)Protein = N% × 6.25

All of the values were reported on a dry matter basis, i.e., weight of the component per total dry weight of the sample (%, (g·100 g^−1^)) (Table 1; Appendix A).

### 2.4. Scanning Samples with Two NIR Instruments

Prior to scanning, the samples were dried at 50 °C for 16 h and cooled to room temperature. The samples were then scanned using a benchtop NIR spectrometer DS2500 flour analyzer from FOSS (FOSS-DS2500; FOSS Electric A/S, Hillerød, Denmark) [5] and Hone Lab’s portable NIR instrument HL-EVT5 (Hone Lab-Engineering Validation Test Model 5; Hone, Newcastle, NSW, Australia) [106].

*Benchtop NIR Instrument:* For obtaining the spectral sample signature from the FOSS-DS2500, each flour sample was transferred to the standard circular ring cup (inside diameter ~6 cm, FOSS sample cup) and scanned three times at room temperature (~26 °C). The sample was mixed before each scan. The NIR spectral absorbance, with a range of 400–2498 nm, was recorded as the logarithm of reciprocal reflectance (1/R) with 2 nm intervals, using the WinISI spectral analytical software (v4.4, InfraSoft International LLC, PA, USA).

*Portable NIR instrument:* To obtain the sample spectral signature from a portable HL-EVT5, the dried flour sample was spread on a glass petri plate with a minimum 5 mm thickness. The instrument was then placed on the layer of flour and scanned at a room temperature of ~26 °C. The instrument was operated via a Bluetooth-connected Hone Create mobile application (v25.2.2 Hone, Newcastle, NSW, Australia; retrieved from play.google.com) (accessed on 8 February 2022). Each sample was scanned at three different points of the sample spread on the petri plate. The mobile application was programmed to record two scans at each position, resulting in six scans per sample. NIR spectra with a range of 1350–2550 nm and a resolution of 16 nm at a wavelength of 1550 nm (NeoSpectra-Micro optical engine, Si-Ware Systems, CA, USA), were extracted from the Hone Create platform [107].

### 2.5. Calibration Model Development

#### 2.5.1. Definition of Calibration and Validation Datasets

The spectral data of 328 samples extracted from the FOSS-DS2500 and the HL-EVT5 were associated with the respective laboratory protein estimates (see Section 2.3). The spectral data from the HL-EVT5 were then split into calibration and validation datasets (80%:20%, respectively). Several methods were considered based on the literature reviews [95,108,109,110,111], but random split was finally made using the Hone Create Platform, as random selection minimizes user bias and is acceptable for large datasets with a normal or uniform distribution. In this case, Hone Create ensures that the minimum and maximum values of protein were designated to the calibration set, and that each species was represented in both the calibration and validation sets, with a ratio of 80:20. Subsequently, the calibration dataset with 262 samples (80% of the total dataset) was used to develop the calibration model and the validation dataset (20% of the total dataset), with 66 samples being used to evaluate the prediction potential of the model (details in Section 2.6). The exact same split of calibration and validation samples was made for the FOSS-DS2500 spectral data. In this way, we formed the basis for evaluating the efficiency of several instruments and the methods of building calibration models—i.e., the WinISI spectral analytical software (v4.4, InfraSoft International LLC, PA, USA), the cloud-based Hone Create software (v25.2.2 Hone, Newcastle, NSW, Australia; retrieved from play.google.com) (accessed on 8 February 2022), and the customized convolution neural network algorithm-based method (TensorFlow/Keras API) [112,113].

#### 2.5.2. Prediction Method Development Using Established Software and a Custom-Made Pipeline: Instrument–Method Combinations

The WinISI analytical software is designed to assess FOSS instrument-generated data in the proprietary data format (.nir). Therefore, the HL-EVT5 instrument data could not be evaluated using the WinISI software. However, the Hone Create Platform enables the users to load any type of data, as long as it is in the prescribed .csv format. Similarly, the customized CNN-based method allows users to import spectral data from any instrument (https://github.com/adamavip/nirs-protein-prediction) (accessed on 20 March 2022). Consequently, it was feasible to treat both spectra types, FOSS-DS2500 and HL-EVT5, using the Hone Create Software, and the customized CNN-based model-building method, while the WinISI could be used only to treat FOSS-DS2500-generated spectra.


*FOSS-DS2500 NIR Spectra Processed using WinISI Software:*


The WinISI software (v4.4) offers several mathematical spectra pre-processing steps: standard normal variate (SNV, range tested), baseline shift, NIR trajectory derivative, and smoothing. After spectral pre-processing, calibrations can be built using several deterministic methods—principal component regression (PCR), partial least squares regression (PLSR), and modified partial least squares (MPLS)—in combination with pre-treatment methods [114,115,116]. Iterations between the methods can be performed manually, and the prediction potential of the models built can be tested using the validation dataset (described in Section 2.5.1). Accordingly, we performed several manual iterations between the available methods. The calibration models achieving the best metrics, i.e., the slope and intercept of linear regression, coefficient of determination (R^2^), root mean squared error (RMSE), and the relative prediction deviation (RPD) for the calibration and validation datasets were then reported.


*FOSS-DS2500 and HL-EVT5 NIR Spectra Processed using Hone Create Software:*


The automated Hone Create software [107] applies a matrix of pre-processing options similar to the WinISI software: baseline correction, area normalization, smoothing, derivative, SNV, or combinations of these techniques. Hone Create automatically iterates and selects the best-performing pre-processing method(s) based on the regression (PLS) or classification (C4.5) models. Once processed, a range of machine learning techniques are automatically tested and compared, including distributed random forest (DRF), generalized linear model (GLM), gradient boosting machine (GBM), extreme gradient boosting (XGBoost), and stacked ensembles. Currently, the best-performing calibration model is selected based on the root mean squared error (RMSE) and the coefficient of determination (R^2^) of the calibration set. “Holdover validation” metrics (i.e., the independent validation set, described in Section 2.5.1.) are automatically processed for the user, with interactive results being displayed to allow the user to interrogate the dataset and to perform further model iterations as needed.

In this study, to compare the prediction potential of instrument–model combinations, the spectra of calibration datasets from the FOSS-DS2500 and the HL-EVT5 instruments (described in Section 2.5.1) were treated separately. The spectral data from the FOSS-DS2500 and the HL-EVT5 in Excel format (.csv) were uploaded into the Hone Create platform. The pipeline was set to automatically iterate the spectrum pre-processing methods before building the calibration model. The best-performing pre-processing method(s) were selected and subsequently used to transform the dataset prior to automatically building the calibration model, using the supervised AutoML framework of Hone Create. Once the optimal calibration model was identified for the calibration set, Hone Create automatically applied the same pre-processing method(s) to the validation set, ran the data against the model, and displayed the analogous metrics, which were then compared with other tested methods (details in Section 2.6).


*FOSS-DS2500 and HL-EVT5 NIR Spectra Processed using a Customized CNN-Based Algorithm:*


So far, more complex ML-algorithms such as convolutional neural networks (CNNs) are difficult to reliably automate within the software interface for regular use by non-experts. Therefore, the methodology involving convolutional neural networks (CNNs) [112] for building multivariate regression calibration models was explored separately. The CNN method was built on the publicly available open-source TensorFlow/Keras API [113]. This CNN was composed of three convolutional layers, three pooling layers, and three fully connected layers. Each convolution layer had 24, 48, and 96 filters, with kernel sizes set to 10, 15, and 25, respectively. All stride parameters were attributed to two. The network was then organized between the convolutional layer and pooling to realize the extraction and mapping of local features from the input NIR dataset. Several fully connected layers were then consecutively arranged, and the regression of targets was performed using a sigmoid function. Batch normalization was added after every convolutional layer to prevent an internal covariate shift and to speed up convergence. The ADAM50 function, a gradient descent algorithm, was set to minimize the loss function with an initial learning rate of 3 × 10^−4^, which enabled the reverse adjustment of weights from the network, using a backpropagation algorithm, reducing the mean squared error of the model after each training iteration [117]. A max-pooling layer, with a kernel filter size set to two, was connected to each layer of activation function. A dropout of 0.02 was then used to deactivate 2% of the network neurons. Finally, the output of the last dropout layer was flattened to represent the high-dimensional features of the input dataset. The extracted high-dimensional features were fed into a multi-layer perceptron (MLP) to execute the final regression task. There were hidden layers in the MLP, with 512 and 128 neurons, successively. A regularization term (index = 10^−7^) was added to every hidden layer to minimize overfitting, followed by batch normalization. The model was trained with a training batch size of 64, using Google Colaboratory (NVIDIA K80s GPU, 12.72 of RAM, and 358.27 GB of hard disk for one runtime), an open-source service provided by Google.

The spectral datasets from the FOSS-DS2500 and the HL-EVT5 were pre-treated using the Savitzky–Golay smoothing filter [114], with a window size of 15 and a polynomial order set to 2, as was performed in similar studies [88,118]. The transformed data, before being fed to the CNN, were then normalized using min/max normalization of the first derivatives so that values ranged between 0 and 1 [37]. Subsequently, the CNN training structure was constructed to predict protein quantity (%, (g·100 g^−1^)) from the spectral data. For this, the calibration set model was trained using a five-fold cross-validation approach to determine the optimal number of epochs and the effectiveness of certain hyperparameters, such as activation functions, neuron counts, and layer counts. The model (with the same architecture as that of the cross-validation) was retrained with the selected hyperparameters on the entire training dataset and tested with the validation dataset (described in Section 2.5.1). Iterations between the pre-treatment and the normalization methods were performed, and the best-performing model was selected based on the common metrics of both the calibration and validation datasets (details in Section 2.6).

### 2.6. Prediction Method Evaluation

To compare the predictive potentials of the different instrument–method combinations, we used the statistical metrics describing the linear interdependency between the ground-truth (grain protein content estimated using the laboratory method, see Section 2.3) and the best method for predicting protein content from the NIR spectra (separately for the calibration and validation sets, see Section 2.5.1). To assess the developed instrument–method combinations (Section 2.5), five parameters were used—the slope and intercept of the linear regression; coefficient of determination (R^2^, Equation (2)); root mean squared error (RMSE, Equation (3)); and the ratio of prediction to deviation (RPD, Equation (4)) [119,120,121],
(2)$$R^{2}\;=\;1\;-\;\frac{\Sigma {\left({\hat{y}}_{i}\;-\;y_{i}\right)}^{2}}{\Sigma {\left({\hat{y}}_{i}\;-\;{\overset{-}{y}}_{i}\right)}^{2}}$$
(3)$$RMSE\;=\;\sqrt{\frac{\Sigma {\left({\hat{y}}_{i}\;-\;y_{i}\right)}^{2}}{n}}$$
where *n* is the number of samples; *y_i_* is the ground-truth (see Section 2.3) value of sample *i**; ŷ_i_* is the model-predicted value of sample *i**;*
*ȳ* is the mean of the ground-truth values; and SD is the standard deviation of the ground-truth values.
(4)$$RPD\;=\;\frac{SD}{RMSE}$$

We adopted the previously-reported classification based on the RPD values [119], wherein an RPD value <1.5 indicates that the calibration is not reliable; a value between 1.5 and 2.0 indicates the capacity of a model to distinguish between high and low values; a value between 2.0 and 2.5 signifies the model’s capacity to “approximate” quantitative prediction; a value between 2.5 and 3.0 suggests “good” quantitative prediction; and a value > 3.0 indicates “excellent” quantitative prediction.

## 3. Results

### 3.1. Diversity of Grain Protein Content in Five Cereal Species

The laboratory analysis of protein content obtained from 328 grain samples across five cereal species ranged from 5.99% to 21.51% (Table 1, Appendix A). The range of protein content in multiple cereals was considerably larger compared to the protein content variability within any of the individual species tested (Table 1, Figure 2). Among the five cereals tested, the mean protein content in pearl millet grains was the highest (21.51%), while the mean protein content was the lowest in finger millet grains (5.99%; Table 1, Figure 2).

Based on the protein content, the Hone Create software randomly splits the samples of each species into calibration and validation datasets (80%:20%, respectively) (Figure 3). The range, average, standard deviation, and distribution of protein content across the calibration and validation datasets were comparable (Appendix A).

### 3.2. NIR Spectrum Obtained from the Benchtop FOSS-DS2500 and the Portable HL-EVT5

The NIR absorbance spectra of 328 samples were recorded using two NIR instruments—the benchtop FOSS-DS2500 (400–2498 nm) and the portable HL-EVT5 (1350–2550 nm). The spectrum profile generated from each instrument was very similar within the range of 1352–2498 nm (Figure 4). This indicated that the technology used to generate the NIR spectral signatures captured the biochemical signature of the biological samples very similarly.

Overall, the FOSS-DS2500 signal was dominated by 13 groups of prominent peaks (Figure 5A) and 5 peaks for the HL-EVT5 (Figure 5B). The protein content is known to be linked to several spectral bands: (i) a range of 950–1050 nm as the N–H second stretch overtone, (ii) around 1500 as the N–H stretching first overtone, and (iii) the N–H bend second overtone, and the C = O stretch–N–H in-plane bending–C–N stretch combination bands are further associated with a range of 2150–2200 nm [1,2,17]. Therefore, both of the instruments should be sufficient to capture some, if not all, of these critical NIR spectral ranges to predict protein content.

### 3.3. NIR Spectrum Generated Using the FOSS-DS2500 and Processed via WinISI Software

The WinISI software enabled manual iterations between the spectra pre-processing steps (SNV, detrend, NIR trajectory derivative, and smoothing) and several deterministic calibration model algorithms (PLS, MPLS, and PCR). For our dataset, the best calibration method that attained the highest accuracy metrics was obtained using spectra pre-processed using combinations of scatter-correction algorithms, SNV&D with a mathematical pre-treatment setting of “2,4,4,1” (i.e., 2 = second derivative treatment; 4= a gap of four wavelength points over which the derivative was calculated; 4 = first smoothing using the Savitzky–Golay algorithm at four data points; and 1 = no secondary smoothing), combined with the modified partial least squares (MPLS) regression. This calibration model achieved RMSE values of 0.91 and an R^2^ of 0.90. The validation dataset possessed RMSE values of 1.09 and an R^2^ of 0.86 (Table 2; Figure 6). The RPD values for the calibration and validation datasets were 3.56 and 3.08, respectively (Table 2).

### 3.4. NIR Spectrum Generated Using the FOSS-DS2500 and HL-EVT5, Processed via Hone Create Software

The Hone Create software is able to perform several combinations of pre-processing methods specific for the generated data and iterate these with a range of deterministic and ML-based calibration methods. The pipeline returns a “Top 10” calibration model leaderboard, with the best-performing calibrations (based on the R^2^ and RMSE of the calibration set).

In the case of the longer spectrum signatures generated using the benchtop FOSS-DS2500 instrument (Figure 6), Hone Create’s best model, achieved with the spectrum transformed using the first-order derivative (pre-processing) combined with the stacked ensemble method, determined the protein content with RMSE values of 0.66 and 1.00, and R^2^ values of 0.96 and 0.90 for the calibration and validation datasets, respectively. This model had an RPD value of 3.38 for the validation and 4.93 for the calibration dataset (Table 2; Appendix A).

For the portable NIR-instrument, HL-EVT5, we observed that the best prediction method was achieved with a spectrum that was preprocessed via area normalization (wherein the magnitude of each value in the spectra is adjusted so that the sum of every absolute magnitude equals 1), followed by spectrum merging steps using smoothing (a Savitsky–Golay filter with a period of 5 and a polynomial order of 2) and a first-order derivative combined with stacked ensemble models (Figure 6). The model showed an R^2^ of 0.98, an RMSE of 0.42, and a RPD of 7.79 for the calibration dataset. The corresponding metrics for the validation set were an R^2^ of 0.91, RMSE of 0.97, and an RPD of 3.48 (Table 2; Appendix A).

### 3.5. NIR Spectrum Generated Using FOSS-DS2500 and HL-EVT5, Processed via CNN-Based Customized Pipeline

The CNN models were also experimented with, for predicting the protein content in cereal grains. The model was built using spectral data preprocessed using the Savitzky–Golay filter, followed by min/max normalization of the first derivatives and a customized deep learning CNN algorithm for model building. For the FOSS-DS2500-generated spectra, the CNN model achieved an R^2^ of 0.99 and an RMSE of 0.33 in the calibration set, and an R^2^ of 0.89 and an RMSE of 1.03 in the validation set (Figure 6, Table 2). Consequently, for the HL-EVT5 instrument samples, an RMSE of 0.46 and 1.10 and an R^2^ of 0.98 and 0.87 were obtained for the calibration and validation datasets, respectively (Figure 6, Table 2). The RPD values for the FOSS-DS2500 and HL-EVT5 instrument-derived validation datasets were 3.26 and 3.06, respectively (Table 2).

### 3.6. Instrument–Method Combination Comparisons for Protein Content Predictions in Cereal Grains

Two instruments (FOSS-DS2500 and HL-EVT5) were used to generate the NIR spectra (Figure 4) of the ground grain samples, and three analytical methods (WinISI software, Hone Create software, and the CNN-based customized algorithm) were used to build the prediction model for grain protein content (%, (g·100 g^−1^)). Five metrics were used to assess the performance of the different methods (slope and intercept of the linear regression, R^2^, RMSE, and RPD; Section 2.6).

Overall, as shown in Table 2, the NIR spectral signals generated from both instruments (FOSS-DS2500 and HL-EVT5) yielded reliable models (the lowest achieved R^2^ ≥ 0.86 and highest RMSE ≤ 1.10 in the validation set). The validation dataset RPD of all of the instrument–method combinations for estimating protein were greater than 3.06 (Table 2). This suggests that all of the generated methods were well suited for providing quantitative protein estimates.

Nevertheless, the prediction methods for the NIR spectra from FOSS-DS2500 using the ML methods achieved a notably higher RPD (≥3.26), a marginally higher R^2^ (≥0.89), and a slightly lower RMSE (≤1.03), compared to the deterministic models created through the WinISI software (RPD = 3.08, R^2^ = 0.86, and RMSE = 1.09) (Table 2). The best-performing instrument–method combination, based on the validation set metrics, was achieved with the HL-EVT5 and Hone Create software.

## 4. Discussion

### 4.1. Importance of NIR Spectroscopy for Rapid Cereal Grain Quality Assessment

The cereals investigated within this study significantly influence the food and nutritional security of farming communities across the tropics [77,78,79,122]. The protein content in these cereals is one of the key parameters determining nutritional grain values in human diets and its suitability for food industries [15,16,17,18,25]. Currently, NIRS-based methods are used for the rapid prediction of organic grain components [15,16,17,18,40,41,42,43,44,45,46,47,48,56,57,58,59,63,64,65,66,89,90,92,93,94]. However, there are not many examples for minor cereal grains (such as millets in the presented study [89,90,123]). We argue that the rapid assessment of minor cereal qualities could be an entry point for their integration into mainstream food-value chains. Therefore, in this work, we assessed the suitability of some of the standard and novel NIR instruments, and of some of the standard and novel model-building methods for the estimation of protein content in minor cereals.

### 4.2. Expected Data Properties as Prerequisites to Building Reliable Prediction Models

Important considerations for building reliable prediction models from the NIR spectra have been comprehensively summarized in [24,110,120,121,123]. In our study, we focused on the following aspects: (i) the number of samples, range of variation and the distribution of the protein content values; (ii) the properties of the spectra generated using two sensors (FOSS-DS2500 and HL-EVT5); and (iv) the methods for building the algorithms. The number, range, and distribution of target trait variability (i.e., protein) are critical prerequisites for the development of reliable calibration models from NIR-based signals [108,120]. Generally, a higher number of precisely generated ground-truth data points increases the probability of building reliable calibrations. While deterministic methods require ~100 ground-truth points, larger sample numbers (>300) are generally required as a basis for ML-based algorithms and industrial calibrations [39,108,124,125]. In our study, we used 328 ground-truth samples, which is an adequate sample size that suits the purpose of the presented study. Additionally, the inclusion of five cereal species extended the range of protein content variability compared to the range found in any of the individual species. Also, both of the instruments tested in this study (FOSS-DS2500, with a spectral range of 400–2498 nm, and HL-EVT5, with a spectral range of 1350–2550 nm) encompassed the spectrum ranges relevant to protein determination in grain samples (Section 3.2) [2,26,28,48,54,65,69,84,91], and sufficiently justified the next step, i.e., building the algorithms to infer the protein content from these two types of spectra (Section 2.5 and 4.3).

### 4.3. Prediction Methods for the Estimation of Protein Content in Multiple Cereal Grains and Their Accuracy Metrics

For the predictions of organic grain composition from NIR spectral reflectance, deterministic methods have been widely used (e.g., MLR, PCR, and PLS regression [15,16,17,18,19,20,25]). These methods were mostly specific to a single species [26,27,28,29,30,31,44,45,46,47,48,49,50,56,57,58,59,60,61,62,63]. The adoption of ML methods in NIR spectroscopy research has quite recently begun, together with the increasing availability of computing power and efficient learning algorithms [34,37,118]. Therefore, in our work, we wanted to test whether the ML algorithms would provide any advantage, in terms of accuracy, to the prediction methods. For this, we tested the portable NIR instrument (HL-EVT5) and the benchtop NIR instrument (FOSS-DS2500) in combination with two software (standard FOSS-made WinISI and novel Hone-made Hone Create), and one externally built CNN-based algorithm. The winISI software leaves the user to manually test the combinations of several preprocessing methods, which proved to be time consuming. In comparison, the Hone Create software enables the automatic evaluation of a similar range of pre-processing methods and models in a relatively shorter amount of time. The one potential constraint of the current Hone Create pipeline is that it selects the “best” models based on the metrics of the calibration dataset. This process might prioritize models that are more specific for the presented dataset, with less generic-prediction capacity (i.e., “over-fitting models”; one sign of overfitting could be where the calibration model metrics are vastly higher compared to the metrics of the validation dataset [88]). This was also one of the reasons for why we tested the alternative process, i.e., the custom-designed CNN pipeline, where we integrated the element where the goodness of the model was evaluated based on the metrics of the validation set. Another reason for building a separate pipeline was that, at this moment, more complex ML-methods such as CNN are difficult to automate. The code is now available on the GitHub platform (https://github.com/adamavip/nirs-protein-prediction, accessed on 23 February 2022), and its particular elements can be utilized to enhance existing software products and to develop other pipelines.

For longer NIR spectral signatures generated using FOSS-DS2500, we compared all three model-building methods (WinISI software, Hone Create software, and custom-made algorithms involving CNN). In this case, it was notable that the prediction methods integrating ML-based models (Section 3.4), particularly the stacked ensemble model via Hone Create (R^2^ = 0.90, RMSE = 1.0, RPD = 3.38) and the custom-designed CNN (R^2^ = 0.89, RMSE = 1.03, RPD = 3.26), achieved better comparative metrics than the deterministic method through WinISI (R^2^ = 0.86, RMSE = 1.09, RPD = 3.08). Overall, all five of the presented calibration models developed for protein assessment can be used for the high-quality quantitative estimation of protein over a range of cereal grains. In addition, the study suggests that the robustness of these calibrations can be further improved by including more diverse samples to further widen the range of trait and spectral variability. Similar studies have been carried out to evaluate the grain protein contents of individual major cereals [26,28,48,54,65]. These studies achieved the accuracy metrics of prediction methods (typically, R^2^ ≥ 0.86; RPD > 3.0) that were comparable to all of the prediction methods presented in this study.

## 5. Conclusions

The integration of minor cereals in mainstream diets will be an important step towards the improvement of human nutrition, as these cereals generally have higher nutritional values compared to the major cereals such as wheat, rice, or maize. We argue that rapid assessment of the minor cereal qualities could be an entry point for their integration into mainstream food-value chains. This will probably require the transition of standard NIR spectroscopic instrumentation for grain quality analysis from the benchtop to the portable form. Such a transition appears to be critical for its effective utilization in decentralized systems where these minor cereals are typically produced. The extended utilization of minor cereal grains by food industries might, in turn, become an important means for the improvement of human nutrition, particularly in the tropics.

The motivation of this study was to assess whether emerging technological approaches (portable NIR instruments and ML-based methods) would enable the accurate assessment of grain composition (protein content) in multiple minor cereal staples that are typical of such agri-food systems. We demonstrated that the NIR spectra generated from a novel portable NIR instrument (HL-EVT5) were sufficient for the reliable quantification of grain protein content in multiple cereal species. The results also show that the integration of ML-based algorithms in modeling processes enhances model accuracy compared to classical deterministic methods. Finally, we highlighted that some of these advanced data-modeling methods are available for non-experts through new software packages (e.g., Hone Create software). We argue that these novel technologies, which are becoming more accessible across all markets, have the power to streamline the production and trade of nutrition-dense food sources (such as millets) into human diets.

## Figures and Tables

**Figure 1 sensors-22-03710-f001:**
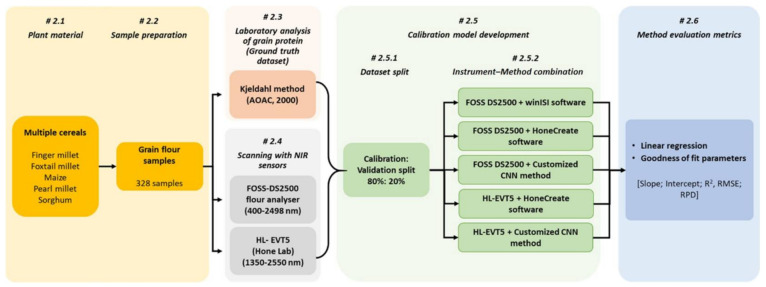
Graphical overview of the methodology visualizing the process used for testing the NIR instruments and methods for prediction of protein content in multiple cereal grains.

**Figure 2 sensors-22-03710-f002:**
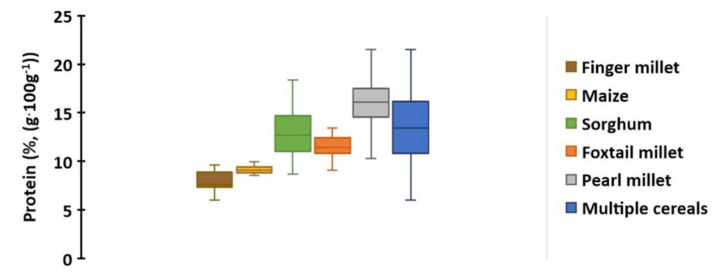
Box plots depicting variation and distribution of protein content (%, (g·100 g^−1^)) in the grains of five cereals, as estimated through laboratory analyses. Legend: Each box represents one crop species; different crops are distinguished by color (finger millet = brown; maize = yellow; sorghum = green; foxtail millet = orange; pearl millet = grey; and the entire set of 328 multiple cereals = blue); solid line within the box (–) represents the mean of each crop.

**Figure 3 sensors-22-03710-f003:**
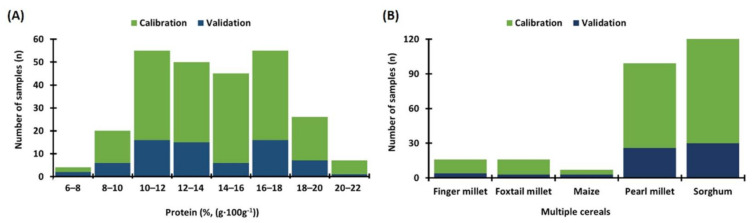
Histograms depicting the distribution of (**A**) the protein content (%, (g·100 g^−1^)) in samples, and (**B**) the number of samples used within each of the crop species belonging to the calibration (80%) and validation (20%) datasets.

**Figure 4 sensors-22-03710-f004:**
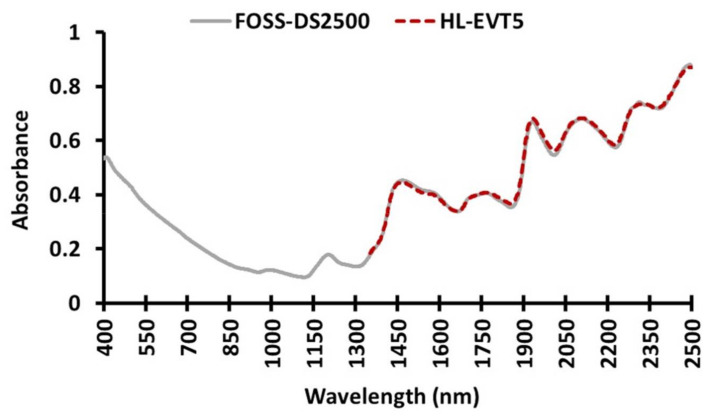
Mean of the near-infrared (NIR) spectra of all grain samples extracted from the benchtop FOSS-DS2500 (400–2498 nm; solid line (–) in grey colour) and the portable HL-EVT5 (1350–2550 nm; dashed line (---) in red colour) instruments.

**Figure 5 sensors-22-03710-f005:**
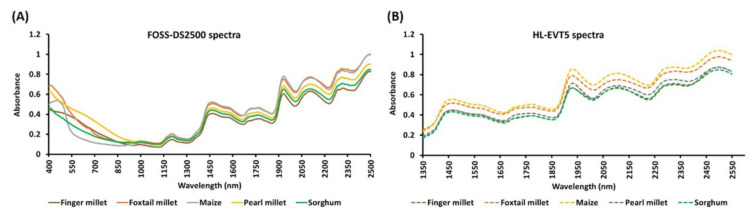
Means of the near-infrared (NIR) spectra of the grain samples of five cereal species produced using (**A**) FOSS-DS2500, 400–2498 nm; solid line (–), and (**B**) HL-EVT5, 1350–2550 nm; dashed line (---) instruments. Different crops are distinguished by color (Legend: finger millet = brown; foxtail millet = orange; maize = yellow; pearl millet = grey; sorghum = green).

**Figure 6 sensors-22-03710-f006:**
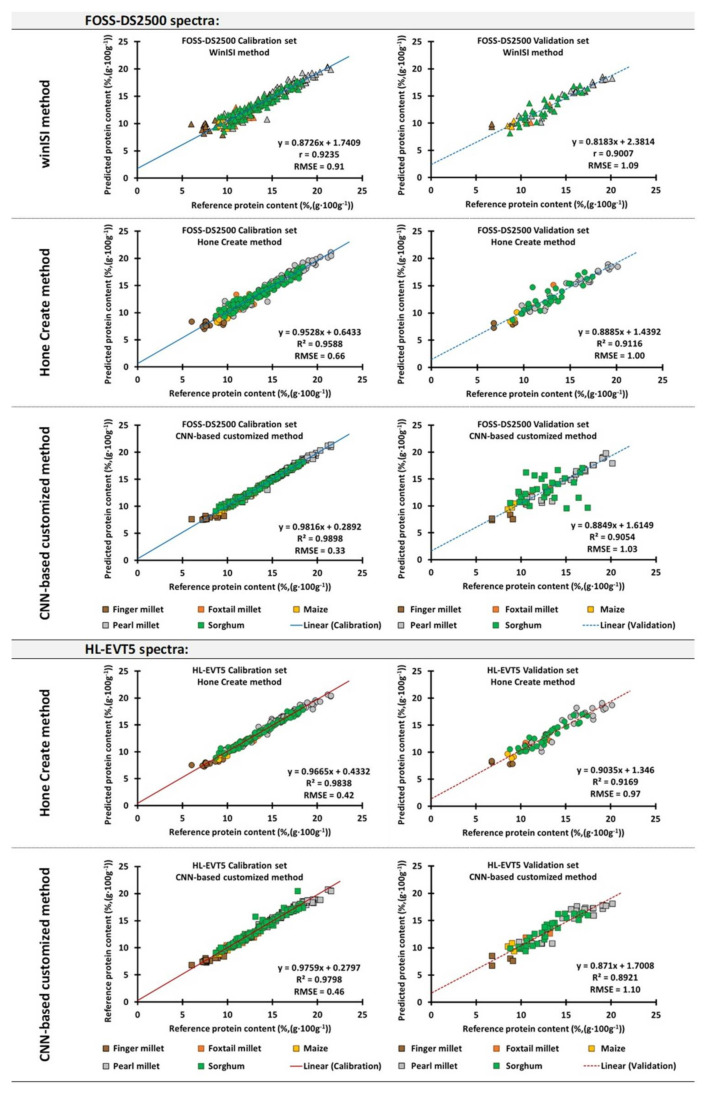
Matrix of scatter plots showing protein predicted for the calibration and validation datasets of FOSS-DS2500 and HL-EVT5 via methods available in (I) WinISI software, (II) Hone Create soft-ware, and (III) CNN-based customized method. Detailed metrics for comparison with other methods are shown in Table 2.

**Table 1 sensors-22-03710-t001:** Results of laboratory estimation of protein content in multiple cereal samples used in the study. The table indicates the range and average grain protein content %, g·100 g^−1^), along with the number of samples used per species.

Species	Number of Samples	Range of Protein (%, (g·100 g^−1^))	Average of Protein (%, (g·100 g^−1^))
Finger millet	20	5.99–9.59	7.93
Foxtail millet	19	9.08–13.42	11.50
Maize	10	8.53–9.97	9.14
Pearl millet	125	9.69–21.51	15.78
Sorghum	154	8.68–18.38	13.09
Multiple cereals	328	5.99–21.51	13.59

**Table 2 sensors-22-03710-t002:** Comparative metrics of NIR spectroscopy calibration (80%) and validation (20%) models developed using combinations of two different instruments (FOSS-DS2500 and HL-EVT5) and three model-building methods (WinISI software, Hone Create software, CNN-based customized pipeline) for protein content estimation in grains of multiple cereal species. Legend: R^2^ = coefficient of determination; RMSE = Root Mean Squared Errors, RPD = ratio of prediction to deviation, CNN = convolutional neural networks.

Instrument	Method	Set	Slope	Intercept	R^2^	RMSE	RPD
FOSS-DS2500	WinISI software	Calibration	0.87	1.74	0.90	0.91	3.56
		Validation	0.82	2.38	0.86	1.09	3.08
	Hone Create software	Calibration	0.95	0.64	0.96	0.66	4.93
		Validation	0.89	1.44	0.90	1.00	3.38
	CNN-based customized pipeline	Calibration	0.98	0.29	0.99	0.33	9.85
		Validation	0.88	1.61	0.89	1.03	3.26
HL-EVT5	Hone Create software	Calibration	0.97	0.43	0.98	0.42	7.79
		Validation	0.90	1.35	0.91	0.97	3.48
	CNN- based customized pipeline	Calibration	0.98	0.28	0.98	0.46	7.00
		Validation	0.87	1.70	0.87	1.10	3.06

## Data Availability

The data related to the laboratory analysis presented in this study is openly available in the Appendix A section of the manuscript. The source code of the CNN-based customized pipeline for estimating protein content in multiple grain cereals using NIRS and machine learning has been published in the Github repository https://github.com/adamavip/nirs-protein-prediction (accessed on 9 April 2022). The spectral data and the model developed using the software winISI and Hone Create presented in this study are not publicly available due to their proprietary nature.

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
