# Peer review of "NIR Instruments and Prediction Methods for Rapid Access to Grain Protein Content in Multiple Cereals"

_sensors, 2022, doi:10.3390/s22103710_

Round 1

Reviewer 1 Report

Dear Author(s)

Please find my comments on the attached file. I think so that discussion part could be written more efficiently. Other parts of manuscript are well written. 

Reviewer 2 Report

Comments

The current research focuses on “Machine learning-powered models for near-infrared spectrometers: prediction of protein in multiple grain cereals”. The work's concept is sensible and interesting; the article can be published after minor revisions based on the suggestions below.

The author should specify the objectives of the study in the introduction part rather than just writing only generalized objectives? Better to write in points?

Authors should add complete detail for all the methods used in the manuscript? Also, by citing the latest relevant literature

Which models give optimum results, should be presented in abstract

Which preprocessing technique was fund suitable for model building with optimum output

Reviewer 3 Report

A major question of the study is to define its aims. The authors used three platforms and three different regression methods to predict the protein content of samples based on their NIR spectra. Additionally, they used different data pretreatment do run the models.

Drawing conclusions from such a diverse run of algorithms is difficult and, to my opinion, is heavily biased.

I see the following possible solutions here:

-compare the two NIR devices but use the same data pretreatments and models

-compare the data pretreatments but use the same models

-compare the models but use the same data pretreatments

-compare the software but use the same data pretreatment and models

My major concern with the paper is the lack of the basis of the comparisons done.
